# Research on the Mechanical Efficiency of High-Speed 2D Piston Pumps

**Yu Huang, Jian Ruan, Chenchen Zhang, Chuan Ding**  **and Sheng Li ***

Key Laboratory of Special Purpose Equipment and Advanced Manufacturing Technology,
Ministry of Education & Zhejiang Province, Zhejiang University of Technology, Hangzhou 310014, China;
YuHuang1117@outlook.com (Y.H.); ruanjiane@zjut.edu.cn (J.R.); chen877698298@outlook.com (C.Z.);
chuanding@zjut.edu.cn (C.D.)
* Correspondence: lishengjx@zjut.edu.cn

**Abstract:** Since many studies on axial piston pumps aim at enhancing their high power-weight ratio, many researchers have focused on the generated mechanical losses by the three friction pairs in such pumps and attempted to diminish them through abundant and new structural designs of the pump's components. In this paper, a high-speed 2D piston pump is introduced and its architecture is specifically described. Afterward, a mathematical model is established to study the pump's mechanical efficiency, including the mechanical losses caused by the viscosity and stirring oil. Additionally, in this study the influences of the rotational speed, the different load pressures, and the rolling friction coefficient between the cone roller and the guiding rail are considered and discussed. By building a test rig, a series of experiments were carried out to prove that the mechanical efficiency was accurately predicted by this model at low load pressures. However, there was an increasing difference between the test results and the analytical outcomes at high pressures. Nevertheless, it is still reasonable to conclude that the rolling friction coefficient changes as the load pressure increases, which leads to a major decrease in the mechanical efficiency in experiments.

**Keywords:** 2D piston machine; mechanical losses; force analysis; mathematical model; experimental research

## 1. Introduction

Axial piston pumps are widely used in many fields because of their high power density, strong load capacity, and long service life [1–3]. Since the invention of axial piston pumps, many designs of their mechanical structure have been innovated by researchers worldwide [4–6]. However, these structural designs have not been applied in a breakthrough, as the mechanical efficiency of axial piston pumps is restricted by the three friction pairs of the cylinder block and valve plate, the cylinder block and pistons, and the slippers and swash plate [7–9].

The team of the author of this paper has for a long time been devoted to designing new hydraulic components using the two-dimensional (2D) concept, which allows the critical parts of hydraulic components, such as the spool in valves or the piston in piston pumps, to have two working degrees of freedom. The 2D concept was successfully applied to various hydraulic components, such as the 2D vibration exciter, 2D servo valves, 2D flowmeters, and 2D piston pumps [10–12]. Traditional 2D piston pumps were previously proposed by linking the 2D designed principle with axial piston pumps [13,14]. Figure 1 shows a schematic diagram of the traditional 2D piston.

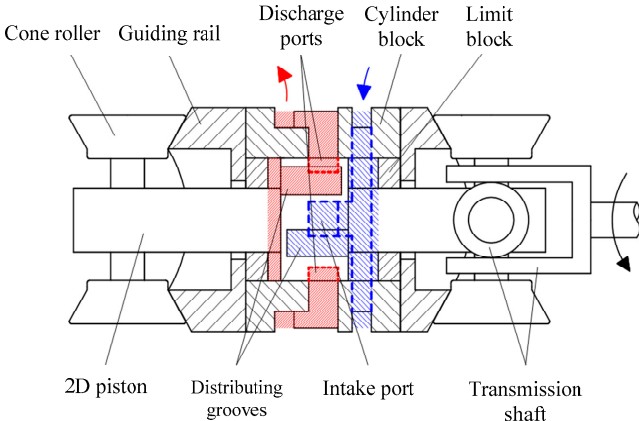

**Figure 1.** Mechanical structure of the traditional 2D piston pump.

The core structure of the 2D piston unit consists of a 2D piston with two pairs of cone rollers fixed on its rods and a cylinder block conjoined with two guiding rails. Two intake ports and two discharge ports are alternatively and circumferentially distributed on the cylinder block. Correspondingly, four distributing grooves are evenly set on the 2D piston and alternatively connected to its left and right sides. The two sides of the 2D piston, the limit blocks, and the cylinder block form two displacement chambers. When the shaft is driven by the motor, the piston's rotational motion is transformed into a reciprocating motion with the help of the combination of the cone roller and the guiding rail, which indicates that the 2D piston reciprocates to change the size of the two displacement chambers, and that it simultaneously rotates to alternatively connect the two displacement chambers to the intake and discharge ports [13]. The above functions are operated by one 2D piston. More specifically, one 2D piston can intake and discharge hydraulic oil four times with each rotation, which is why traditional 2D piston pumps are more compact than normal axial piston pumps and are characterized by a high power-weight ratio.

Increasing the rotational speed is one of the most important methods for enhancing the power-weight ratio of axial piston pumps. For example, the rotational speed of the axial piston pump used in the F-35 fighter jet exceeds 10,000 rpm [15,16]. This rule is also suitable when it is applied to the 2D piston pump. However, the unbalanced inertial force from a single 2D piston would result in the vibration of the cylinder block at high rotational speeds, which is a shortcoming of traditional 2D piston pumps when attempting to gain even higher power-weight ratios.

In order to adapt a high rotational speed input, a novel high-speed 2D piston pump is proposed to eliminate the unbalanced inertial force, as shown in Figure 2. The critical idea is adding a balancing set to balance the inertial force which is generated by the driving set. More specifically, as in Figure 3a,b, the driving set includes a 2D piston with two pairs of driving rollers fixed on its rods, whereas the balance set has two pairs of balancing rollers that are solidly connected to each other through a transmission shaft. The left driving roller and left balancing roller are combined as a cross, and they roll on the left guiding rail, as illustrated in Figure 3c. The right roller set is similarly and symmetrically designed just as the left roller set, as shown in Figure 2. Therefore, the two displacement chambers are composed of the 2D piston of the driving set, the piston rings of the balancing set, and the cylinder block. The design of the other parts of the novel high-speed 2D piston pump, such as the grooves, windows, and the profiled curve of the rails, obeys the same principle as the traditional 2D piston pump.

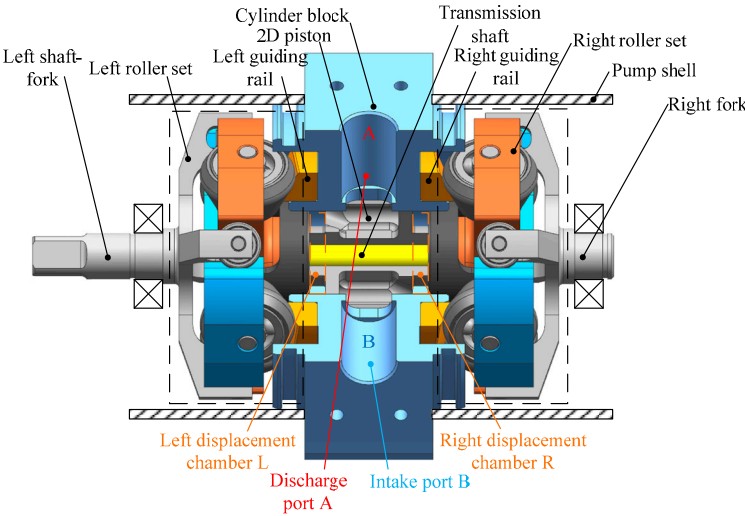

**Figure 2.** Mechanical structure of the high-speed 2D piston pump [17].

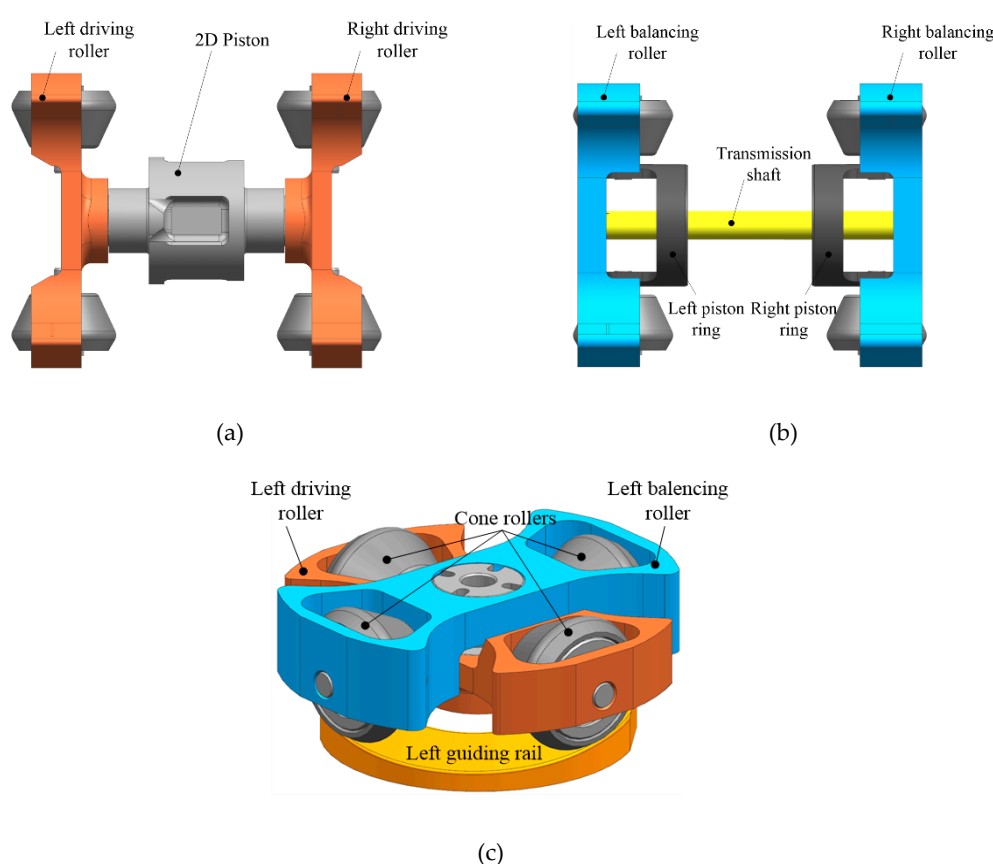

**Figure 3.** Critical structures of the high-speed 2D piston pump [17]. (**a**) Driving set; (**b**) balancing set; (**c**) left roller set.

The fork shaft is connected to the driving motor through a coupling, and it drives the two roller sets when the pump starts to work. Since the two roller sets are constrained by their own guiding rails, the 2D piston and the piston rings are forced to reciprocate. However, unlike the rotational motion in which they rotate at the same pace, the reciprocating motion of the piston rings has a 90-degree phase difference with the reciprocating motion of the 2D piston due to the motion design, as shown in Figure 4a. As a result, it always compresses one displacement chamber and opens another, thus mapping the intake and discharge processes of the pump, respectively. The theoretical output

flow rate of the high-speed 2D piston pump is shown in Figure 4b. The relative motion of the 2D piston and the piston rings leads to the most compact design ever. In addition, since the two rollers work on one guiding rail, the relative displacement is doubled with the stroke of the guiding rail, which means that the output flow is also doubled, leading to the enhancement of the power-weight ratio.

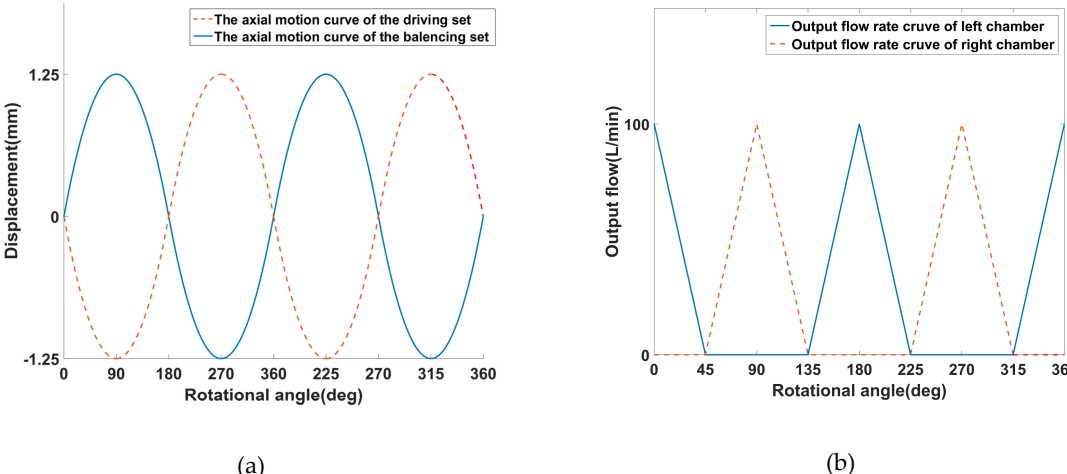

(a)                                    (b)

**Figure 4.** Theoretical curves of the high-speed 2D piston pump [17]. (**a**) The curves of the reciprocating motions; (**b**) the curves of the output flow rates.

Various studies have been carried out on the high-speed 2D piston pump. Jin obtained a design method of the surface of the guiding rail through a mathematical model and verified it using experiments [13]. Shentu analyzed the flow and pressure ripples through commercial computational fluid dynamics (CFD) software and obtained the volumetric efficiency [18]. Huang calculated the generated churning torque losses using the driving and balancing sets at high rotational speeds via CFD numerical calculation and proposed a theoretical formula that is consistent with the experimental data, especially when the rotational speed is below 8000 rpm [17].

However, the mechanical efficiency of the high-speed 2D piston pump, as a critical characteristic for every pump, has not been well studied yet. This paper presents a specific mathematical modeling of the mechanical efficiency of the high-speed 2D piston pump. First, the force analysis of the high-speed 2D piston pump is carried out to lay the foundation stone. Then, the churning loss and viscosity damping loss are considered in the mathematical model. Afterward, the influences of three aspects, which are the rotational speed, load pressure, and rolling friction coefficient, are considered and analyzed in order to figure out the reason leading to the increasing difference between the analytical and experimental results.

## 2. Mathematical Model

Due to the similarity between the balancing set and the driving set, their load conditions are based on the same principle, which is specifically described by taking the driving set as an example. According to the displacement curves in Figure 4, the whole period of one rotation can be divided into four intervals, where each interval has a 90-degree rotation and one complete stroke in the axial direction. Afterward, due to the direction of acceleration, each interval has two different situations that are required to be separately analyzed.

Assuming the state of the pump described in Figure 5 as the initial analysis time in this paper, the state of the pump is at the maximum volume of the left displacement chamber and the first situation, where, in the first situation, the direction of acceleration is toward the displacement chamber with the pressurized hydraulic oil as the 2D piston rotates from 0 degree to 45 degrees, as shown by the red dash line from 45 degrees to 90 degrees in Figure 4a. Figure 5 demonstrates the force analysis of the first situation. Since a uniform acceleration and deceleration rule is applied to the reciprocating motion

design, the driving set moves to the left with a uniform acceleration [13]. Meanwhile, the hydraulic oil in the left displacement chamber is depressed to discharge, which leads to a hydraulic pressure force, $F_p$, on the 2D piston. Besides this, an axial shear force, $F_{sh}$, on the gap, $h$, which is between the 2D piston and the cylinder block, is also presented as resistance due to the axial motion.

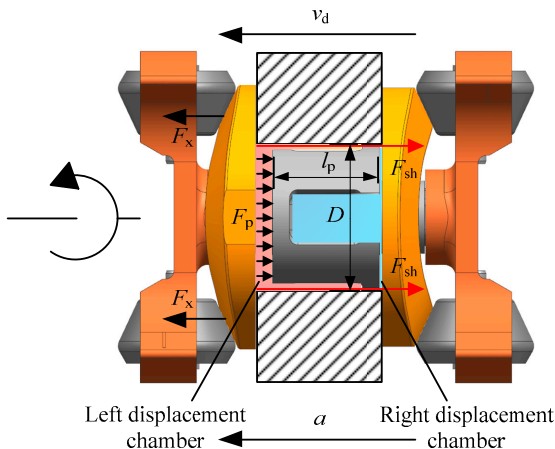

**Figure 5.** Force analysis of the driving set while rotating from 0 degrees to 45 degrees.

In order to overcome the above two forces, the axial driving force, $F_x$, on the driving set is supported by the left guiding rail, and, according to Newton's second law, the force balance equation can be described by Equation (1).

$$ma = F_x - \left(F_p + F_{sh}\right) = F_x - \left(p_i A_p + \frac{\mu D \pi l_p v_x}{h}\right), \tag{1}$$

where $m$ is the mass of the driving set, $p_i$ is the pressure in the left displacement chamber, $A_p$ is the cross-sectional area of the 2D piston, $\mu$ is the oil dynamic viscosity, $D$ is the diameter of the 2D piston, $l_p$ is the length of the 2D piston, and $v_x$ is the reciprocation velocity referenced from Equation (2) from our previous research [18].

$$v_x = \begin{cases} at & 0 < t \le \frac{\pi}{4\omega} \\ a\left(\frac{\pi}{2\omega} - t\right) & \frac{\pi}{4\omega} < t \le \frac{3\pi}{4\omega} \\ a\left(t - \frac{\pi}{\omega}\right) & \frac{3\pi}{4\omega} < t \le \frac{\pi}{\omega} \end{cases}, \tag{2}$$

where $a$ is the acceleration based on the rotational speed $n$ and 2D piston stroke $L_s$, and $\omega$ is the rotational angular velocity.

In order to obtain the axial driving force, it is necessary to carry out a force analysis on the contact between the cone rollers and the guiding rail. As shown in Figure 6a, the supporting force of the guiding rail for the cone roller, $F_s$, is perpendicular to the contact line of the guiding rail. The angle between the axial component force of the supporting force, $F_{sx}$, and the supporting force is half of the cone angle, $\theta_c$.

When the cone rollers roll on the guiding rail, they receive the rolling friction force, $F_f$, which is proportional to the supporting force with a rolling friction coefficient. Therefore, the axial driving force is composed of the axial component forces of both the supporting force and the rolling friction force, which can be described by Equation (3).

$$F_s = \frac{F_x}{\cos\left(\frac{\theta_c}{2}\right)\cos(\theta_p) - \mu_f \sin(\theta_p)}, \tag{3}$$

where $\mu_f$ is the rolling friction coefficient between the cone roller rolls and the guiding rail, and $\theta_p$ is pressure angle between the cone roller and the guiding rail, which has been well studied and can be expressed by Equation (4) [13].

$$\theta_p = arctan\left(\left|\frac{v_X}{\omega R_r}\right|\right), \tag{4}$$

where $R_r$ is the radius of the driving set.

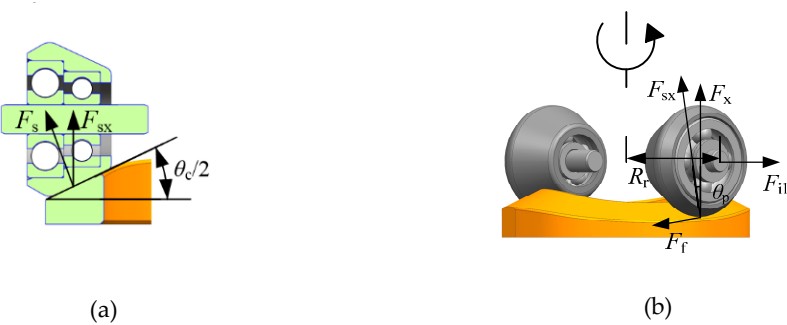

(a)                                                        (b)

**Figure 6.** Force analysis of the cone rollers on the left driving set when the driving set rotates from 0 degrees to 45 degrees. (**a**) Axial component force of the supporting force on the cone roller; (**b**) force analysis of the cone rollers on the guiding rail.

Assuming that the rotational speed of the input motor is constant, during this interval the circumferential torque of the supporting force, $T_{i1}$, can be obtained using Equation (5).

$$T_{i1} = F_{i1}\cdot R_r = F_s\cdot\left(cos\left(\frac{\theta_c}{2}\right)sin(\theta_p) + \mu_f cos(\theta_p)\right)\cdot R_r, \tag{5}$$

where $F_{i1}$ is the driving force that drives the driving set to rotate.

The second situation of the interval is much more complicated because of the constant acceleration design when the 2D piston rotates from 45 degrees to 90 degrees. In this situation, the direction of acceleration is the same as that of the direction of the force from the pressurized hydraulic oil. Therefore, the axial force balance equation can be expressed through the force analysis of Figure 7.

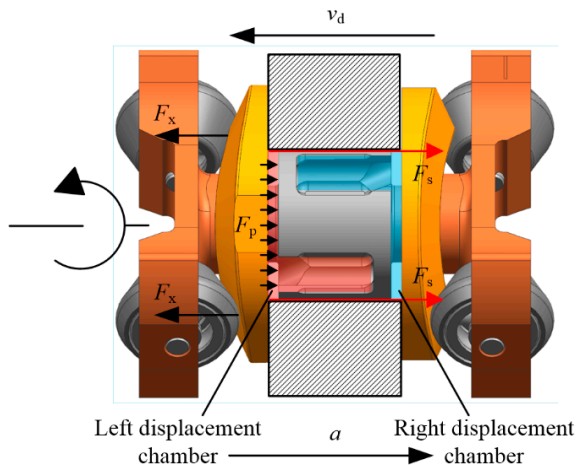

**Figure 7.** Force analysis of the driving set when rotating from 45 degrees to 90 degrees.

Since the acceleration is constant and is estimated through the guiding rail profile, the product of the mass and the acceleration is also constant. Hereby, the axial driving force makes the driving set decelerate with the same acceleration using the left guiding rail. The equations of the input torque are consistent with Equation (5), and the description of the pressure angle is consistent with Equation (4).

However, when the hydraulic pressure force is relatively low, the axial driving force is provided using the right guiding rail, as demonstrated in Figure 8. In this case, the forces between the cone rollers and the right guiding rail are shown in Figure 9 and described by Equations (6)–(8), respectively.

$$ma = \left(F_{\mathrm{p}} + F_{\mathrm{sh}}\right) - F_{\mathrm{x}}, \tag{6}$$

$$F_{\mathrm{s}} = \frac{F_{\mathrm{x}}}{\cos(\frac{\theta_{\mathrm{c}}}{2})\cos(\theta_{\mathrm{p}}) + \mu_{\mathrm{f}}sin(\theta_{\mathrm{p}})}, \tag{7}$$

$$T_{\mathrm{i1}} = F_{\mathrm{i1}} \cdot R_{\mathrm{r}} = F_{\mathrm{s}} \cdot \left(\cos(\frac{\theta_{\mathrm{c}}}{2})sin(\theta_{\mathrm{p}}) - \mu_{\mathrm{f}}cos(\theta_{\mathrm{p}})\right) \cdot R_{\mathrm{r}}. \tag{8}$$

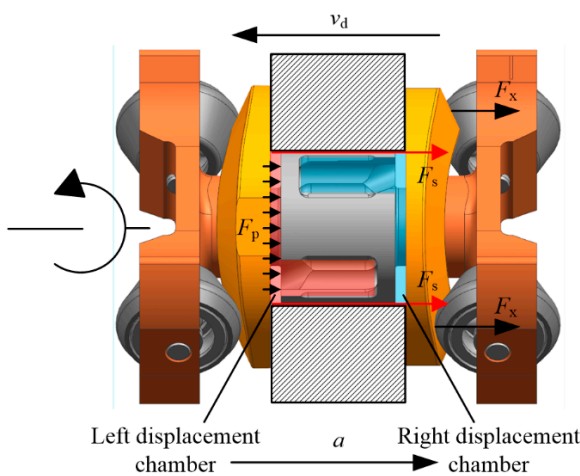

**Figure 8.** Force analysis of the driving set while rotating from 45 degrees to 90 degrees when the load pressure is low.

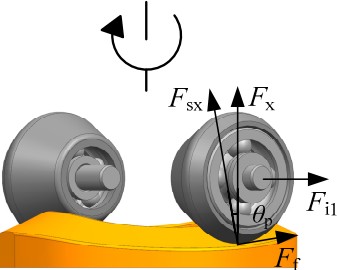

**Figure 9.** Force analysis of the cone rollers on the right driving set when it rotates from 45 degrees to 90 degrees.

In addition, two sorts of torque occur in the high-speed 2D piston pump due to the rotation of the 2D pistons. The first one is the shearing torque, $T_{\mathrm{s}}$, which results from the flow in the gaps between the pistons and the cylinders, as shown in Figure 10, and it can be obtained using Equation (9).

$$T_{\mathrm{s}} = \frac{\mu D\pi\left(l_{\mathrm{p}} + 2l_{\mathrm{pr}}\right)\omega\frac{D}{2}}{h}\frac{D}{2}, \tag{9}$$

where $l_{\mathrm{pr}}$ is the width of the piston rings.

The other torque is caused by the cone rollers of both the driving and balancing sets, which are rotating in oil, and it is considered as the churning losses torque, $T_{\mathrm{c}}$. A careful research has been previously carried out to establish a mathematical model for evaluating the influence of the churning losses torque [17]. As a result, the obtained methodology can be referenced here, and Equation (10) can be used to describe the churning losses torque.

$$T_{\text{c}} = 7.25 \times 10^{-5} \cdot n - 2.5 \times 10^{-9} \cdot n^2, \tag{10}$$

where the unity of the rotational speed is rpm, and the unity of the churning losses torque is Nm.

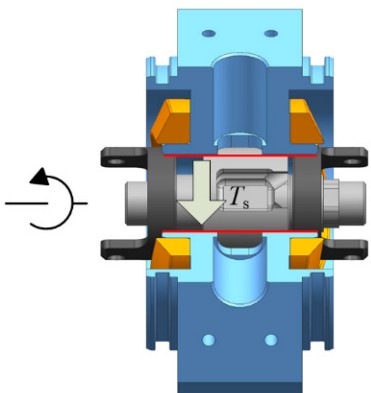

**Figure 10.** Resistance torque caused by the shearing flow during rotation.

In conclusion, when the driving and balancing sets rotate from 0 degrees to 90 degrees, the input torque of the motor is $T_{\text{ai}}$, and it can be obtained using Equation (11).

$$T_{\text{ai}} = T_{\text{i1}} + T_{\text{i2}} + T_{\text{s}} + T_{\text{c}}, \tag{11}$$

where $T_{\text{i2}}$ is the input torque of the balancing set in which the same basic rule applies, and it is unnecessary to repeat the derivation process here.

Therefore, the mechanical efficiency, $\eta$, of this pump can be calculated using Equation (12).

$$\eta = \frac{p_{\text{i}} V_{\text{m}} / (2\pi)}{\dfrac{\int_0^{t_{\text{90deg}}} (T_{\text{i1}} + T_{\text{i2}}) dt}{t_{\text{90deg}}} + T_{\text{s}} + T_{\text{c}}}, \tag{12}$$

where $t_{\text{90deg}}$ is the time in which the sets rotate from 0 degrees to 90 degrees, and $V_{\text{m}}$ is the displacement of the pump, where its unity is m³/rotation.

## 3. Theoretical Analysis

Based on the above mathematic model, the influences of the rotational speed, load pressure, and rolling friction coefficient on the analytical results are sequentially studied in this section. In order to simplify the calculations, the pressure loss caused by the fluid tunnels and the throttle loss between the windows and the slots were both neglected, which means that the load pressure was assumed to be equal to the pressure of the pressurized displacement chamber. The values of the relevant parameters are shown in Table 1.

### 3.1. Rotational Speed

In traditional axial piston pumps, it is mainly considered that various rotational speeds normally affect the mechanical losses generated by friction pairs [15]. However, in high-speed 2D piston pumps, the increase in the rotational speeds results in an increase in both the reciprocation speed and the rotational speed of two active sets, which enlarges the mechanical losses of the stirring oil and the mechanical losses caused by the viscosity damping, both axial and circumferential.

When the load pressure is assumed as 8 MPa and keeps at a constant value, the analytical input torques of the driving set are as shown in Figure 11 when the rotational speed increases from 1000 to 5000 rpm. Since the load pressure is constant, the difference among the curves is caused by the above

two velocity-sensitive factors, except for the moment at the rotational angle of 45 degrees. At this sudden moment, cliff changes occur, and their amplitudes increase with the increase in the rotational speed, which is because of the instantaneous alternation of the acceleration direction. The input torques have to overcome the inertia of the active components, which increases with the increase in the rotational speeds.

**Table 1.** Parameters of the mathematical model.

| Description | Value |
| --- | --- |
| Rotational speed $n$ | 5000 rpm |
| Load pressure | 8 MPa |
| Rolling friction coefficient $\mu_f$ | 0.001 |
| Oil dynamic viscosity $\mu$ | 0.038930 Pa·s |
| Gap between 2D piston and cylinder block $h$ | $2 \times 10^{-5}$ m |
| Mass of the driving set or the balancing set $m$ | 0.1 kg |
| Motion stroke | 0.0025 m |
| Acceleration $a$ | $1.11 \times 10^3$ m/s$^2$ |
| Cone angle of the cone roller $\theta_c$ | 60 degrees |
| Diameter of the 2D piston $D$ | 0.01275 m |
| Cross-sectional area of the 2D piston $A_p$ | $2.56 \times 10^{-4}$ m$^2$ |
| Length of the 2D piston $l_p$ | 0.02 m |
| Radius of driving set $R_r$ | 0.02 m |
| Width of piston rings $l_{pr}$ | 0.015 m |
| pump's displacement $V_m$ | $5.12 \times 10^{-6}$ m$^3$/rotation |

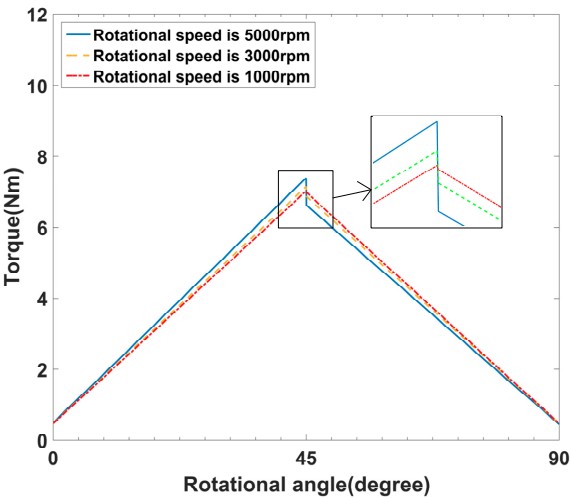

**Figure 11.** The input torques of the driving set at different rotational speeds and at a load pressure of 8 MPa.

Figure 12 shows that the amplitudes of the input torques as the 45-degree rotational angle decreases with the reduction in the piston mass, which verifies the above analysis.

The obtained theoretical mechanical efficiencies are shown in Figure 13b with various increasing rotational speeds. It can be obviously observed that the theoretical mechanical efficiency decreased with the increase in the rotational speed. More specifically, as in Figure 13a, the basic mechanical efficiency slightly decreased as the rotational speed increased. Meanwhile, the theoretical mechanical efficiencies with either the viscosity damping or churning loss significantly decreased when the rotational speed was high. The way to minimize the energy loss from the viscosity damping is to enlarge the gap between the piston and the cylinder, which would lead to a larger internal leakage and, thus, a lower volumetric efficiency. However, it is still worthwhile to find an optimized point between the two efficiencies. The energy loss from the churning losses can be relatively easy to handle

by carrying an optimization process on the shapes of the driving and balancing sets through CFD numerical calculations.

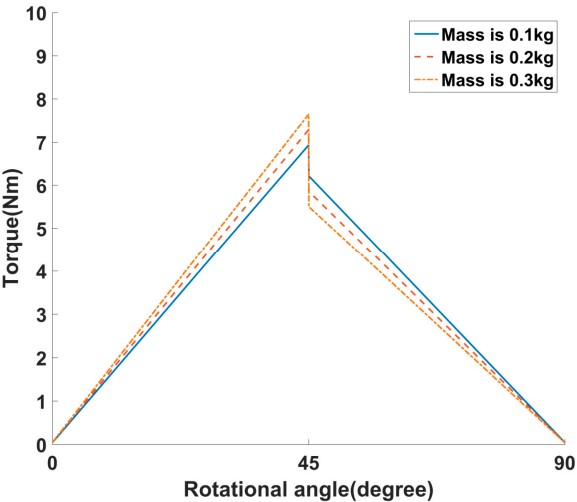

**Figure 12.** The input torques of the driving set at different masses.

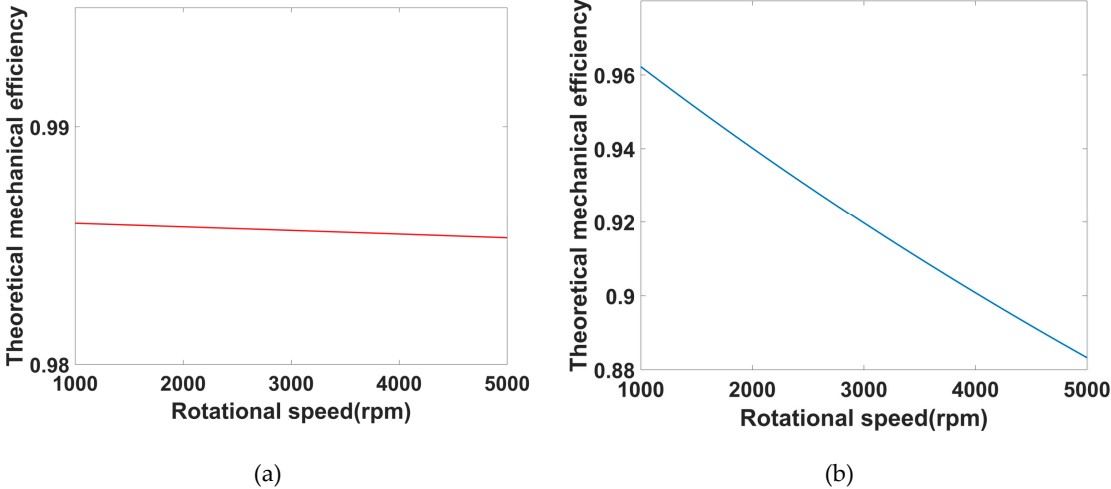

(a)　　　　　　　　　　　　　　　　　　　　　　　　　　　(b)

**Figure 13.** Theoretical mechanical efficiencies at different rotational speeds. (**a**) Without considering the viscosity damping and oil stirring losses; (**b**) with considering the viscosity damping and oil stirring losses.

### 3.2. Load Pressure

The load pressure effects on the mechanical efficiency are mainly reflected in the rolling friction force. As the load pressure increases, the supporting force of the guiding rail for the cone rollers increases, thus resulting in the enlargement of the rolling friction force and eventually leading to an increase in the mechanical losses.

When the driving set rotates from 0 degrees to 90 degrees and the rotational speed is constant, the input torque is strengthened as the load pressure increases, as shown in Figure 14a. Additionally, as shown in Figure 14b, it should be noted that there is a unique phenomenon in the high-speed 2D piston pump, as the calculated input torque can be negative when the driving set rotates from 45 degrees to 90 degrees. This is due to the design of the uniform acceleration deceleration axial motions for 2D pistons, and, as described in Figure 8 in the previous section, the acceleration is maintained by the support of the right guiding rail. Figure 15 demonstrates the relationship between the theoretical mechanical efficiency and the increase in the load pressures.

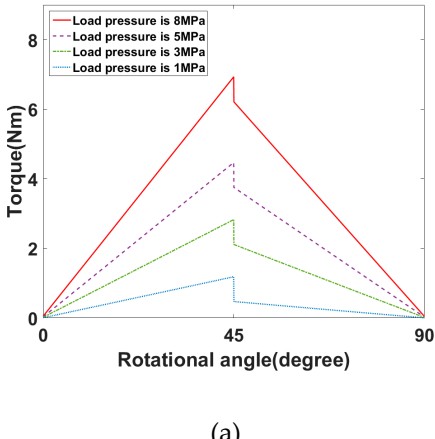

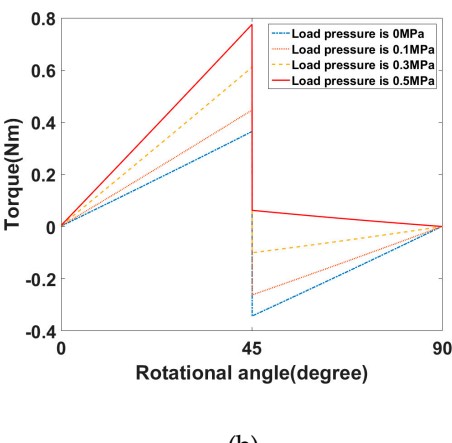

(a) (b)

**Figure 14.** Input torque distribution of the driving set under different load pressures at 5000 rpm. (**a**) Normal load pressures; (**b**) relatively low load pressures.

According to Equation (12), the load pressure itself can dramatically influence the calculation of the mechanical efficiency, which indicates that the trend in Figure 15 is logical.

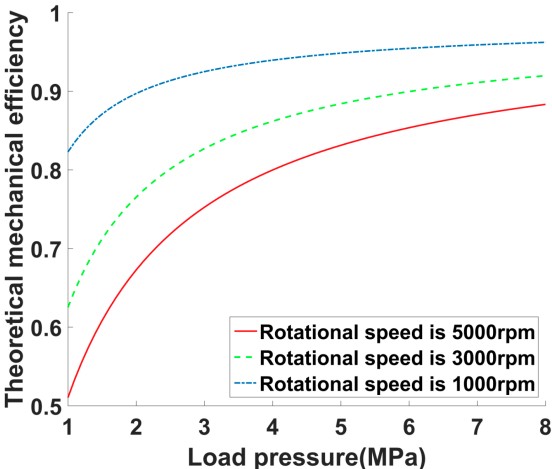

**Figure 15.** Theoretical mechanical efficiency under different load pressures at 5000 rpm.

## 4. Experimental Research

To verify the mathematical model and the theoretical analysis, an experimental rig, as shown in Figure 16, was established to measure the mechanical efficiency of the high-speed 2D piston pump. A supply pump was installed at the inlet of the tested pump to ensure that the sucking oil of the high-speed 2D piston pump was sufficient. The used driving motor for the high-speed 2D piston pump was a 30 kW three-phase asynchronous electric motor with a maximum speed of 20,000 rpm. Flexible couplings were installed at both ends of the torque/speed sensor to ensure the stability of the transmission chain at high rotational speeds. Additionally, in order to accurately measure the load pressure, pressure sensors were installed at the inlet and outlet of the tested pump. The details of the relevant sensors are described in Table 2.

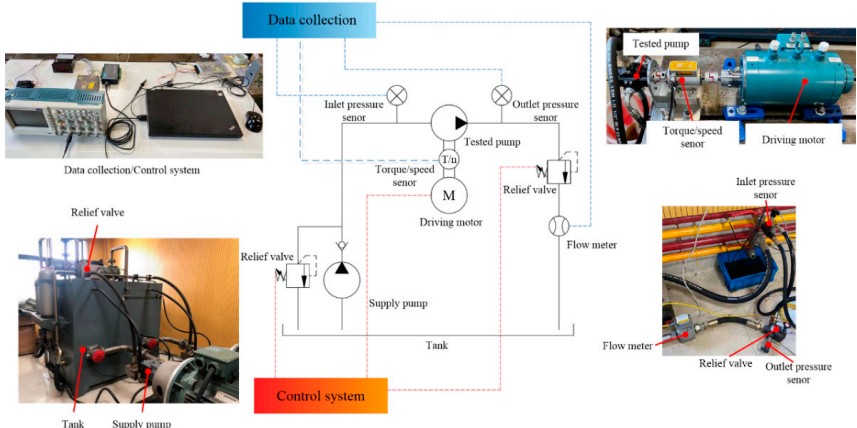

**Figure 16.** Experimental schematic diagram.

**Table 2.** Details of the relevant sensors.

| Description | Accuracy |
|---|---|
| Torque/Speed sensor | Torque range 0–20 Nm; accuracy ±0.1%; rotational speed range 0–18,000 rpm |
| Pressure sensor | Range 0–100 bar, accuracy ±0.3% |

The experimental and analytical results are shown in Figure 17. As shown in the figure, the experimental results of the mechanical efficiency of the tested pump are consistent with the analytical results at a low load pressure. However, when the load pressure increases, the differences between the experimental and analytical results significantly increase. Meanwhile, the influence of the rotational speeds on the difference is not that apparent. The turning points of the experimental results are 3 MPa at 1000 rpm and 4 MPa at 5000 rpm, respectively.

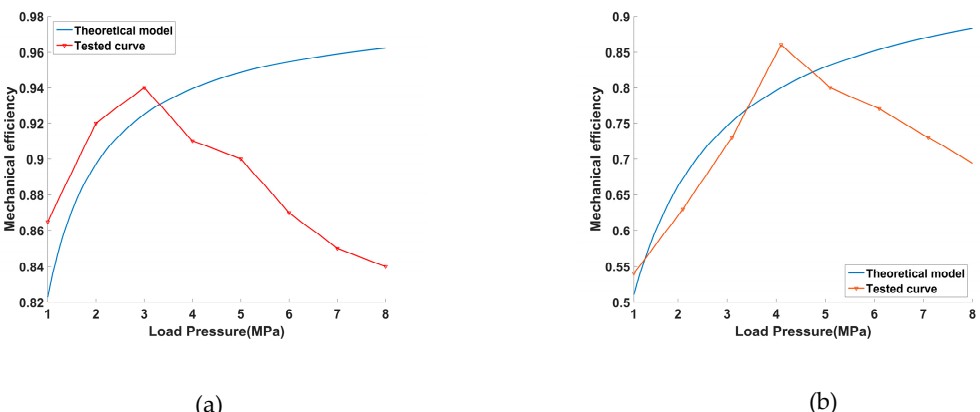

(a)    (b)

**Figure 17.** Comparison between the experimental and analytical results of the high-speed 2D piston pump. (**a**) At 1000 rpm; (**b**) at 5000 rpm.

The reason for this large difference between the experimental and analytical results might be due to the change in the rolling friction coefficient when the load pressure increases. A study that focused on the experimental observation of the mechanical efficiency of axial piston pumps demonstrated that the axial pump has a similar trend when the load pressure is high [19]. This is because the working states of the three sliding friction pairs in axial pumps change, which leads to higher mechanical losses.

It is reasonable to speculate that in the experiment carried out on the high-speed 2D piston pump, the same situation happened with the rolling friction coefficient between the cone rollers and the guiding rail. When the load pressure increased, the rolling friction state changed, resulting in a change in the rolling friction coefficient and thus affecting the mechanical efficiency of the pump [20].

Therefore, Figure 18 illustrates the analytical results with different rolling friction coefficients when the rotational speed was at 5000 rpm. In comparison with the experimental results, the rolling friction coefficient was 0.01 when the load pressure was below 6 MPa, and it increased to 0.02 when the load pressure was below 8 MPa.

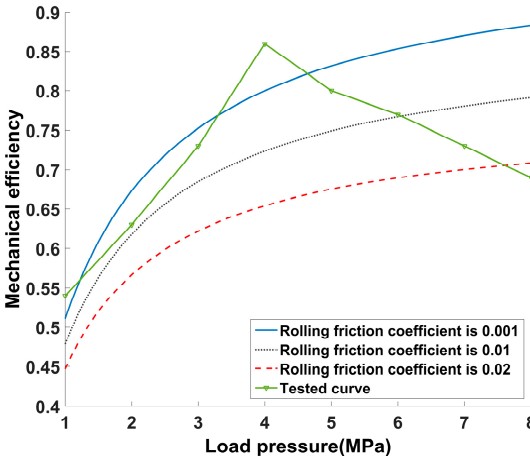

**Figure 18.** Comparison between the experimental and analytical results at different rolling friction coefficients at 5000 rpm.

## 5. Conclusions

First, this paper describes the force analysis of the high-speed 2D piston pump and establishes a mathematical model to predict its mechanical efficiency. Second, the influences of three aspects, which are the rotational speed, load pressure, and rolling friction coefficient, on the mechanical efficiency were analyzed. An experimental rig was built to prove the mathematical model. Finally, the mathematical model and the experimental data were compared, and the differences between them were explained. The following conclusions were drawn.

(1) The high-speed 2D piston pump was affected by the viscosity and stirring oil losses at high rotational speeds. Therefore, reducing the above two losses is an important way to improve the mechanical efficiency.

(2) The mathematical model accurately predicted the mechanical efficiency under different rotational speeds at a low load pressure.

(3) When the load pressure increased, the mechanical efficiency continued to decline. The explanation of the above phenomenon is that as the load pressure increases, the working state of the rolling friction between the cone roller and the guiding rail changes, which decreases the mechanical efficiency.

Future research on the mechanical efficiency will concern the contact between the cone roller and the guiding rail based on the Hertz contact theory. Moreover, the relationship between the rolling friction coefficient, rotational speed, and load pressure will be studied using mathematical modeling. Additionally, the alternation of the working state of the friction pairs in 2D pumps deserves deep research to establish a more accurate mathematical model for predicting the mechanical efficiency of high-speed 2D piston pumps at the full range of load pressures.

**Author Contributions:** Resources, J.R.; Data Curation, C.Z.; Writing—Original Draft Preparation, Y.H.; Writing—Review & Editing, C.D.; Funding Acquisition, S.L. All authors have read and agreed to the published version of the manuscript.

**Funding:** This research was funded by the National Natural Science Foundation of China, grant number 51675482, 51805480, and by the National Key Research and Development Program of China, grant number 2019YFB2005201.

**Conflicts of Interest:** The authors declare no conflict of interest.

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
