# Peer review of "Research on the Mechanical Efficiency of High-Speed 2D Piston Pumps"

_processes, doi:10.3390/pr8070853_

Round 1

Reviewer 1 Report

The Authors focused on a high-speed 2D piston pump and its architecture which has been descripted. The mathematical model has been established by Authors in order to study the mechanical efficiency of the pump, which includes the mechanical losses caused by of the viscosity and the stirring oil. Furthermore, the experimental investigation has been conducted by means of the built up a test rig. The article seems to be interesting but there are some mistakes and they should be corrected. Some of the comments on the manuscript are listed below.

1) The dynamical equation for translational motion (1) is written in a proper way but the equation for rotational motion is not written in the same manner. The Reviewer propose to unify the representation of the equations of motion in the same way in the whole manuscript.

2) Line 22 and 23; some keywords have been already used in the title of the manuscript. Please change them into different ones (to avoid the keywords repetition with the words used in the title).

3) Figure 4. Why is the range of an angle on the abscissa from 0 to 315 degrees and not from 0 to 360 degrees what corresponds to the full cycle?

4) If the displacement of the reciprocating motion represents sinusoidal function then the velocity should be represented by the cosinusoidal function. If yes, why does the volumetric flow rate represent the triangular course?

5) Why is the mass of the liquid neglected in equation (1) and no information is given to the Readers?

6) The Authors should give information to the Readers about the mass of the pump and the mass of the of the oil occupying the pump and etc.

7) Equation 2, 3, 10, 11 and 12; if the equations are taken from the literature the appropriate citations should be given to the Readers.

8) How the Authors understand the rolling coefficient of friction because in the literature there are two different kinds of interpretations.

9) Could you check the units in equation (9) because from the Reviewer’s point of view they are not correct?

10) The font of the captions in the figures could be bigger (comparable with the size of the font of the text).

11) Table 1; the unit of the oil dynamic viscosity is wrong. Could you specify which coefficient of viscosity do Authors use in their manuscript (kinematic or dynamic)?

Furthermore there are some mistakes in units for example an acceleration should be in m/s^2 not as it is shown in the mentioned table in "m^2/s".

The standard computer notation (e.g. 2.56e-4) should be changed into scientific notation (e.g. 2.56·10-4).

Author Response

Reviewer#1, Concern # 1: The dynamical equation for translational motion (1) is written in a proper way but the equation for rotational motion is not written in the same manner. The Reviewer propose to unify the representation of the equations of motion in the same way in the whole manuscript.

Author response:  Thanks for your comment. We have written “However, unlike the rotation motion in which they rotate at the same pace, the reciprocating motion of the piston rings has 90 degrees phase difference with the reciprocating motion of 2D piston due to the motion design in figure 4a.” in line 74 and the rotational speed is constant. So we think rotational equation can be neglected.

Author action:

Reviewer#1, Concern # 2: Line 22 and 23; some keywords have been already used in the title of the manuscript. Please change them into different ones (to avoid the keywords repetition with the words used in the title).

Author response:  Thanks for your comment. We have corrected it

Author action: We have corrected it. The keywords: 2D piston machine; Mechanical losses; Force analysis; Mathematical model; Experimental research

Reviewer#1, Concern # 3: Figure 4. Why is the range of an angle on the abscissa from 0 to 315 degrees and not from 0 to 360 degrees what corresponds to the full cycle?

Author response: Thanks for your comment. We will add new picture from 0 to 360 degrees.

Author action: We have corrected it

Reviewer#1, Concern # 4: If the displacement of the reciprocating motion represents sinusoidal function then the velocity should be represented by the cosinusoidal function. If yes, why does the volumetric flow rate represent the triangular course?

Author response: Thanks for your comment. The displacement of the reciprocating motion is composed of quadratic curves, which looks like sinusoidal. However, it follows the law of uniform acceleration and deceleration curve which is applied on the design of the guiding rail. Because the flow is obtained by the formula Q=A*v, where A is cross area and constant, the curve of the flow is consisted with the triangular course.

Author action:

Reviewer#1, Concern # 5: Why is the mass of the liquid neglected in equation (1) and no information is given to the Readers?

Author response: Thanks for your comment. The liquid hinders the movement of the 2D piston by pressure, but its own mass is normally not counted in a hydraulic system.

Author action:

Reviewer#1, Concern # 6: The Authors should give information to the Readers about the mass of the pump and the mass of the of the oil occupying the pump and etc.

Author response: Thanks for your comment. We have given the mass of driving and balancing sets in the simulation parameters that are necessary for simulating. The oil density is 870 kg/m3, but it is not used in the mathematical model.

Author action:

Reviewer#1, Concern # 7: Equation 2, 3, 10, 11 and 12; if the equations are taken from the literature the appropriate citations should be given to the Readers.

Author response: Thanks for your comment. We have noted the cited literature [18] for Equation 2 and the cited literature [17] for Equation 10. Equation 3 and 11 are ordinary force analysis. Equation 12 is described in detail.

Author action:

Reviewer#1, Concern # 8: How the Authors understand the rolling coefficient of friction because in the literature there are two different kinds of interpretations.

Author response: Thanks for your comment. We have written “When the cone rollers roll on the guiding rail, they receive the rolling friction force, Ff, which is proportional to the supporting force with a rolling friction coefficient.” in line 131.  In order to understand, we abbreviate it as “the rolling friction coefficient between the cone roller rolls and the guiding rail”. The meaning of the two is the same.

Author action:

Reviewer#1, Concern # 9: Could you check the units in equation (9) because from the Reviewer’s point of view they are not correct?

Author response: Thanks for your comment.  is the shear force formula under a laminar flow.

Author action:

Reviewer#1, Concern # 10: The font of the captions in the figures could be bigger (comparable with the size of the font of the text).

Author response: Thanks for your comment. We have adjusted the font size.

Author action:

Reviewer#1, Concern # 11: Table 1; the unit of the oil dynamic viscosity is wrong. Could you specify which coefficient of viscosity do Authors use in their manuscript (kinematic or dynamic)?

Furthermore there are some mistakes in units for example an acceleration should be in m/s^2 not as it is shown in the mentioned table in "m^2/s".

The standard computer notation (e.g. 2.56e-4) should be changed into scientific notation (e.g. 2.56·10-4).

Author response: Thanks for your comment. We have corrected it.

Author action:

Table 1. Parameters of mathematical model

Description

Value

Rotational speed n

5,000 rpm

Load pressure

8 MPa

Rolling friction coefficient μf

0.001

Oil kinematic viscosity μ

0.038930 m2/s

Gap between 2D piston and cylinder block h

2·10-5 m

Mass of the driving set or the balancing set m

0.1 kg

Motion stroke

0.0025 m

Acceleration a

1.11·103 m­/s2

Cone angle of the cone roller θc

60 deg

Diameter of the 2D piston D

0.01275 m

Cross-sectional area of the 2D piston Ap

2.56·10-4 m2

Length of the 2D piston lp

0.02 m

Radius of driving set Rr

0.02 m

Width of piston rings lpr

0.015 m

pump’s displacement Vm

5.12·10-63/rotation

Reviewer 2 Report

Dear authors,

Is interesting your manuscript and can be improved.

I included some comments and observations as follows. 

The manuscript must be improved and must found the friction source for break-down of the efficiency over a given pressure.

Best regards,

Reviewer

Details:

equation (1): Obs.1 Is not clearly this sign! If we accept the Newton low for viscous friction Fsh = tau*A = miu*(vx/h)*A. Is not indicate the film thickness h! 

equation (5): Obs 2.:Please explain the equation of the force Fi1!

equation (8): Similar comments like Obs.2! 

equation (9): Obs. 4: Please indicate if this sign is film thickness!

equation (10): Obs.5: In Eq. 10 it is necessary to explain the unity of n ( probably rpm!) and the unity of the Torque Tc (Nm or Nmm !) because is an empirical equation! 

equation (12):

Obs.6: The Torques Ti1 and Ti2 are expressed in Nm. If propose an integration according to time the integral will be expressed in Nm second! Can be summed with the other torques Ts + Tc (Expresed in Nm)?

Obs.7: The torques Ti1 and Ti2 are depending on the time (t)? Can you present the dependences between Ti1, Ti2 and time?

Line 170:

form should be from;

Obs.8 I think is the volumetric debit of the oil!

Table 1:

0.001: Obs. 8. I think is a very low value!

2e-5 m: Obs.9 : Please correct! is Pas as unity for dynamic viscosity and not mm^2/s ( kinematic viscosity!)

Acceleration a: Obs.10: What means this value for acceleration?

5.12e-6 m3/r: Obs.11: Probably m^3/rotation?

Line 245-246: Comments 1: I propose that will be better that instead a constant rolling friction to be introduced a rolling friction moment MER as function of material, diameter and normal load and to realize the equilibrium of the forces and moments on each roller. Is a suggestion. The rolling friction coefficient can not be at this very low value of o,oo1 

Figure 18: Comment 2: The important break down of the efficiency at a given pressure suggest existence of an important friction source developed over a given pressure. According to the Fig. 18 is not only the rolling friction coefficient. Must be detect this friction source.

Author Response

Reviewer#2, Concern # 1 and 4: Obs.1 Is not clearly this sign! If we accept the Newton low for viscous friction Fsh = tau*A = miu*(vx/h)*A.

Is not indicate the film thickness h!

Author response: Thanks for your comment. We have corrected it. h is the gap between the 2D piston and the cylinder block.   We can find the same formula in formula 2 in the following reference

“Zhang,J.H.,Li,Y.,Xu,B.,Pan,M.,&Lv,F.(2017).Experimentalstudyontheinfluenceoftherotatingcylinderblockand pistons on churning losses in axial piston pumps.Energies, 10(5),662.https://doi.org/10.3390/en10050662”

Author action: We modify line115 “Besides, an axial shear force, Fsh, on the gap, h, between the 2D piston and the cylinder block is also presented as a resistance due to the axial motion.” and equation1

Reviewer#2, Concern # 2 and 3: Please explain the equation of the  force Fi1!

Author response: Thanks for your comment. We will explain the force Fi1.

Author action: We add “where Fi1 is the driving force that drives the driving set to rotate.” in line 141.

Reviewer#2, Concern # 5: In Eq. 10 it is necessary to explain the unity of n ( probably rpm!) and the unity of the Torque Tc (Nm or Nmm !) because is an empirical equation!

Author response: Thanks for your comment. We will add the unity.

Author action: We add “where the unity of the rotational speed is rpm and the unity of the churning losses torque is Nm.”

Reviewer#2, Concern # 6: The Torques Ti1 and Ti2 are expressed in Nm. If propose an integration  according to time the integral will be expressed in Nm second! Can be summed with the other torques Ts + Tc (Expresed in Nm)?

Author response: Thanks for your comment. There is an error in this formula and we will correct it.

Author action: We have modified formula 12

Reviewer#2, Concern # 7: The torques Ti1 and Ti2 are depending on the time (t)? Can you present the dependences between Ti1, Ti2 and time?

Author response: Thanks for your comment. The dependences between Ti1 and time are described in figure 11, 12, and 14.

Author action:

Reviewer#2, Concern # 8: I think is the volumetric debit of the oil!

Author response: Thanks for your comment.  is the equation of the theoretical input torque where Vm is the displacement of the pump and its unity is m3/rotation.

Author action: We have added “Vm is the displacement of the pump and its unity is m3/rotation.” in line 172.

Reviewer#2, Concern # 9: I think is a very low value!

Author response: Thanks for your comment. The steel-to-steel rolling friction coefficient is indeed so small, but this ignores the effect of the load. It can be found from the following URL.

https://www.school-for-champions.com/science/friction_rolling_coefficient.htm#.XwVtted5v-g

Author action:

Reviewer#2, Concern # 10 and 11: Please correct! is Pas as unity for dynamic viscosity and not mm^2/s ( kinematic viscosity!) What means this value for acceleration? Probably m^3/rotation?

Author response: Thanks for your comment. We have corrected it.

Author action:

Reviewer#2, Concern # 12: I propose that will be better that instead a constant rolling friction to be introduced a rolling friction moment  MER as function of material, diameter and normal load and to realize the equilibrium of the forces and moments on each roller. Is a suggestion. The rolling friction coefficient can not be at this very low value of o,oo1

Author response: Thanks for your comment. This idea is consistent with our further research goals, and we will add this part in the conclusion. The steel-to-steel rolling friction coefficient is indeed so small, but this ignores the effect of the load.

Author action: We add “Future research on the mechanical efficiency will concern the contact between the cone roller and the guiding rail based on the Hertz contact theory. Moreover, the relationship between the rolling friction coefficient, rotational speed, and load pressure will be studied using mathematical modeling.” in the conclusion.

Reviewer#2, Concern # 13: The important break down of the efficiency at a given pressure  suggest existence of an important friction source  developed over a given pressure.  According to the Fig. 18 is not only the rolling friction coefficient. Must be detect this friction source.

Author response: Thanks for your comment. In this article, we want to establish a basic mathematical model of the mechanical efficiency of the 2D pump, and improve this mathematical model by adding the churning and viscosity losses. Now we can only explain this difference between experimental results and simulations by increasing the rolling friction coefficient.

As the rolling friction coefficient increases, the intersection of simulation and experimental results continues to move backward. This shows that the increase in the rolling friction coefficient can indeed cause the decrease in mechanical efficiency.

The relationship between the rolling friction coefficient and the load and the study of friction pairs in the 2D pump are the focus of our future research.

But it does not rule out the influence of other friction pairs, we will explain this in the conclusion.

Author action: We add “Also, the alternation of the working state of the friction pairs in 2D pumps deserves deep research to establish a more accurate mathematical model for predicting the mechanical efficiency of high-speed 2D piston pumps at the full range of load pressures.” in the conclusion.

Reviewer 3 Report

The authors present theoretical and numerical results concerning a high-speed 2D friction pump. In the first part, they introduce the design proposed while, in the second part of the paper, they develop a mathematical model that is aimed at determining the overall efficiency of the pump. The paper has it technical significance, less scientific soundness, nevertheless I recommend publication after the following comments are taken into account.

There are few typos which I report here:

line 10. "...by three friction pairs", which one???

line 17. "...experiments is carried..."  

line 72. "The fork shaft is connected to the"

line 115. "In order to overcome above two forces", rephrase, missing "the"???

- Check all the equations. Eq (1) and (9) contain a strange symbol... 

Fig. 13, maybe both the curves in a single panel?

line 144. "...the description curve...", maybe the "guiding rail profile"

line 146. "to maintain the equation..." ?????

Fig. 17 and 18 may be improved, maybe also the model... For example, the authors say that it is the rolling friction coefficient which increases with the velocity. Indded, it could be a possible explanation. What if in their formula they insert a friction coefficient that is dependent on the velocity? Maybe a polynomial or exponential formula... would this improve the prediction also at high speed?

Author Response

Reviewer#3, Concern # 1: line 10. "...by three friction pairs", which one???

Author response: Thanks for your comment. In the introduction, we explained these three friction pairs and indicated references. “However, these structural designs have not been applied in a breakthrough, as the mechanical efficiency of axial piston pumps is restricted by the three friction pairs of the cylinder block and valve plate, the cylinder block and pistons, and the slippers and swash plate [7–9].” In line 29-32.

Author action:

Reviewer#3, Concern # 2: line 17. "...experiments is carried..."  

Author response: Thanks for your comment. We have corrected it.

Author action:

Reviewer#3, Concern # 3: line 72. "The fork shaft is connected to the"

Author response: Thanks for your comment. We have corrected it.

Author action:

Reviewer#3, Concern # 4: line 115. "In order to overcome above two forces", rephrase, missing "the"???

Author response: Thanks for your comment. We have corrected it.

Author action: We have modified to “In order to overcome the above two forces”.

Reviewer#3, Concern # 5: - Check all the equations. Eq (1) and (9) contain a strange symbol

 Author response: Thanks for your comment. We have corrected it.

Author action:

Reviewer#3, Concern # 6: Fig. 13, maybe both the curves in a single panel?

Author response: Thanks for your comment. The theoretical mechanical efficiency without the consideration of the losses from the viscosity damping and the oil stirring is decline as the increase of the rotational speed. This drop is very small and you need to zoom in on the ordinate axis to notice it, so it cannot be placed in the same panel.

Author action:

Reviewer#3, Concern # 7: line 144. "...the description curve...", maybe the "guiding rail profile"

Author response: Thanks for your comment. We have corrected it.

Author action: We have modified to “Since the acceleration is a constant and decided through the guiding rail profile”.

Reviewer#3, Concern # 8: line 146. "to maintain the equation..." ?????

Author response: Thanks for your comment. We have corrected it.

Author action: We have modified to “When the load pressure is high, the hydraulic pressure force is large enough to maintain that the left guiding rail is actioned by the axial driving force.”

Reviewer#3, Concern # 9: Fig. 17 and 18 may be improved, maybe also the model... For example, the authors say that it is the rolling friction coefficient which increases with the velocity. Indeed, it could be a possible explanation. What if in their formula they insert a friction coefficient that is dependent on the velocity? Maybe a polynomial or exponential formula... would this improve the prediction also at high speed?

Author response: Thanks for your comment. We believe that the effect of rotational speed on the rolling friction coefficient is less than the effect of load pressure, so we only mention the effect of load pressure in the article. In further research, we will consider the effect of rotational speed on the rolling friction coefficient.

Author action: We have added “Future research on the mechanical efficiency will concern the contact between the cone roller and the guiding rail based on the Hertz contact theory. Moreover, the relationship between the rolling friction coefficient, rotational speed, and load pressure will be studied using mathematical modeling. Also, the alternation of the working state of the friction pairs in 2D pumps deserves deep research to establish a more accurate mathematical model for predicting the mechanical efficiency of high-speed 2D piston pumps at the full range of load pressures.” in the conclusion.

Round 2

Reviewer 2 Report

Dear authors,

The manuscript has been improved and can be accepted for publication  but some minor corrections must be realized.

Please change the unity of the dynamic viscosity in Pas and not m^2/s and more attentions for the value of acceleration.

In the attached comment I determined the values for speed vx and obtained a very low value.

Best regards

Author Response

Reviewer#2, Concern # 1 and 2:Please change the unity of the dynamic viscosity in Pas and not m^2/s and more attentions for the value of acceleration.

In the attached comment I determined the values for speed vx and obtained a very low value.

Author response: Thanks for your comment. We are very sorry that we did not check the data of the article by ourselves, and we have corrected it now. The the dynamic viscosity is 0.038930 Pa·s and the Acceleration a is  1.11·103 m/s2. The maximum speed of the vx is 1.67m/s.

Reviewer 3 Report

Thanks for having addressed all my concerns.

Author Response

Thank you for your comments